# Investigating Efficiently Extending Transformers for Long Input Summarization

**Jason Phang [1]\*** **Yao Zhao[2]** **Peter J. Liu[2]**
[1]New York University, [2]Google Research, Brain Team
jasonphang@nyu.edu
{yaozhaoyz, peterjliu}@google.com

## Abstract

While large pretrained Transformer models have proven highly capable at tackling natural language tasks, handling long sequence inputs still poses a significant challenge. One such task is long input summarization, where inputs are longer than the maximum input context of most models. Through an extensive set of experiments, we investigate what model architectural changes and pretraining paradigms most efficiently adapt a pretrained Transformer for long input summarization. We find that a staggered, block-local Transformer with global encoder tokens strikes a good balance of performance and efficiency, and that an additional pretraining phase on long sequences meaningfully improves downstream summarization performance. Based on our findings, we introduce PEGASUS-X, an extension of the PEGASUS model with additional long input pretraining to handle inputs of up to 16K tokens, which achieves strong performance on long input summarization tasks comparable with much larger models.

## 1 Introduction

Large pretrained Transformer models have proven to be extremely capable at tackling natural language tasks (Devlin et al., 2018; Brown et al., 2020). However, handling long textual sequences continues to be a significant challenge for these models. Training models to handle long sequences is expensive in both computation and memory, and moreover requires training and evaluating on long sequence data, which is rarer and more costly to collect. Given the broad success of Transformer models on short-sequence language tasks, our goal is to investigate the best way to extend these models to handle longer input sequences.

In this work, we focus on the task of long input summarization: summarizing long input documents into shorter text sequences. The inputs

---
\* Work done while at Google.

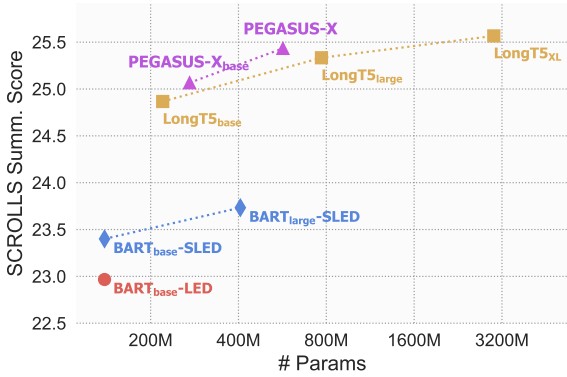

Figure 1: Performance on SCROLLS (Shaham et al., 2022) summarization tasks. All models evaluated on up to 16K input tokens. PEGASUS-X outperforms other models at comparable model sizes. Scores (as of 08/08/22) shown are the average of the geometric mean of ROUGE-1/2/L.

of such tasks are often significantly longer than the maximum input lengths of most standard Transformer models, and hence warrant both architecture modifications as well as new training regimes. For instance, to avoid the quadratic growth in memory consumption of attention in Transformers, many memory-efficient Transformer variants have been proposed (Tay et al., 2020, 2021). However, the manner in which these changes are incorporated into models has been inconsistent and ad-hoc, and there are few established best practices. For instance, some works directly fine-tune on long-input summarization tasks (Zaheer et al., 2020; Pang et al., 2022), while others first perform additional pretraining (Beltagy et al., 2020). Because of the high cost of training these models, there has yet to be a systematic study of how best to adapt models for long input sequences. Hence, it has been difficult to establish which model and training changes are necessary or complementary.

To answer these questions, we conduct an extensive empirical investigation into the architectural changes, model configurations and pretraining

schemes to identify better approaches to training Transformer models for long input summarization. We evaluate a set of efficient Transformer variants, and propose a simple block-wise local Transformer architecture with staggered blocks and global tokens that strikes a good balance of performance and memory efficiency. We show that given a fixed token budget, pretraining on short sequences and then pre-adapting the model to an efficient Transformer architecture by training on longer sequences leads to better performance than only long input pretraining or no adaptation at all. We also investigate model design choices such as position encoding schemes, encoder-decoder layer distributions, and the impact of discrepancies between pretraining and fine-tuning architecture hyperparameters.

Based on the findings from our empirical investigation, we adapt the pretrained PEGASUS$_{\text{Large}}$ model (Zhang et al., 2020) to tackle long input summarization on up to 16K input tokens. The resulting model, which we call PEGASUS-X, attains top scores on long summarization tasks, outperforming much larger models like LongT5 (Guo et al., 2021). Moreover, impact on short input summarization performance is minimal. A smaller version which we call PEGASUS-X$_{\text{Base}}$ attains similar scores with much fewer parameters. Beyond summarization, we believe that many of our findings will be useful to the community for efficiently adapting Transformer models to handle ever longer input sequences for other tasks.

In summary, our contributions are:

1. We evaluate a series of proposed efficient Transformer architectures as well as other model modifications, and report their efficacy and computational trade-offs when applied to long input summarization tasks.

2. Based on our findings, we propose a recipe for adapting a short-context, pretrained Transformer encoder-decoder to longer inputs, and apply it to PEGASUS to greatly improve its long-document summarization performance, with comparable short-input performance.

## 2 Experimental Setup

Similar to Zhang et al. (2020), we perform the majority of our experiments with a PEGASUS$_{\text{Base}}$-sized model, before applying our findings to PEGASUS$_{\text{Large}}$-sized model.

### 2.1 Pretraining

We generally follow the recipe from PEGASUS (Zhang et al., 2020) for pretraining PEGASUS$_{\text{Base}}$-sized models. All experiments in our ablation study performed pretraining with C4 (Raffel et al., 2020) for 500k steps with 512 input tokens and 256 output tokens and a masking ratio of 45%, unless otherwise stated. For long input pretraining we extend the input length to 4096 tokens, and adjust the masking ratio from 45% to 5.625%, reducing the ratio by a factor of 8 to account for the 8x increase in input sequence length. We also filter for only documents longer than 10000 characters.

### 2.2 Fine-tuning

We evaluate models by fine-tuning on the arXiv (Cohan et al., 2018) and GovReport (Huang et al., 2021) long input summarization tasks. Where relevant, we also fine-tune on the shorter-context XSUM and CNN/DailyMail tasks. For each experiment, we report the best validation set scores based on the geometric average (RG) of ROUGE-1, ROUGE-2 and ROUGE-L scores (Lin, 2004) based on the `rouge-score` package.[1] Fine-tuning hyperparameters can be found in Appendix E. Unless otherwise stated, we directly switch to the efficient Transformer architectures between pretraining (on shorter context) and fine-tuning (on longer contexts), with no adaptation phase in between.

## 3 Ablation Experiments

### 3.1 Encoder architectures

We first investigate whether using an efficient Transformer encoder allows models to incorporate longer input sequences while consuming reasonable amounts of device memory. We consider two encoder architectures that exemplify different approaches to efficient attention. Big Bird (Zaheer et al., 2020) uses sparse attention computation, combining sliding-window and random attention, and a set of global-attention tokens. Conversely, Performer (Choromanski et al., 2021) factorizes attention matrices via orthogonal random features. Both model also performed well on the LRA tasks (Tay et al., 2021). For this experiment, we perform both pretraining and fine-tuning with the same encoder architecture to avoid the issue of mismatch between pretraining and fine-tuning architectures.

---

[1] https://github.com/google-research/google-research/tree/master/rouge

| Encoder | XSUM | | CNN/DM | | arXiv | | GovReport | | Steps/s | Mem |
| | R1 / R2 / RL | RG | R1 / R2 / RL | RG | R1 / R2 / RL | RG | R1 / R2 / RL | RG | | |
|---|---|---|---|---|---|---|---|---|---|---|
| Transformer | **40.0 / 16.9 / 32.0** | **27.9** | **39.5 / 19.0 / 28.6** | **27.8** | - / - / - | - | - / - / - | - | - | - |
| BigBird | 39.6 / 16.7 / 31.7 | 27.6 | 39.3 / 18.2 / 28.1 | 27.2 | 46.8 / 19.6 / 28.0 | 29.5 | 60.5 / 28.5 / 30.1 | 37.3 | 0.31 | 1.88 |
| Performer | 36.5 / 14.0 / 28.7 | 24.5 | 37.4 / 17.4 / 26.9 | 26.0 | 39.0 / 13.2 / 23.8 | 23.1 | 55.8 / 20.2 / 24.7 | 30.3 | 0.96 | 1.12 |
| Local | 38.5 / 15.7 / 30.6 | 26.4 | 39.0 / 18.4 / 28.1 | 27.2 | 46.5 / 19.7 / 27.9 | 29.5 | 60.2 / 28.3 / 30.0 | 37.1 | **1.00** | **1.00** |
| Global-Local | 38.7 / 16.2 / 31.2 | 26.9 | 39.0 / 18.6 / 28.2 | 27.3 | **47.6 / 20.2 / 28.5** | **30.1** | 61.4 / 29.3 / 30.6 | 38.0 | 0.87 | 1.08 |

Table 1: Comparison of different encoder architectures on short (XSUM, CNN/DM) and long (arXiv, GovReport) summarization tasks. Training steps per second and memory are computed based on arXiv, and normalized to Local Transformer performance.

| Encoder | Stagger Local Blocks | Use Global In Decoder | arXiv | | GovReport | |
| | | | R1 / R2 / RL | RG | R1 / R2 / RL | RG |
|---|---|---|---|---|---|---|
| Global-Local | ✓ | ✓ | **48.1 / 20.3 / 28.5** | **30.3** | 60.5 / 28.8 / 30.5 | 37.6 |
| Global-Local | | ✓ | 47.0 / 19.5 / 27.9 | 29.5 | 60.9 / 28.9 / 30.2 | 37.6 |
| Global-Local | ✓ | | 47.7 / 20.4 / 28.6 | 30.3 | **61.3 / 29.4 / 30.8** | **38.1** |
| Global-Local | | | 46.7 / 19.5 / 27.9 | 29.4 | 59.5 / 27.8 / 29.4 | 36.5 |
| Local | ✓ | - | 46.8 / 19.7 / 28.0 | 29.6 | 59.2 / 27.9 / 30.0 | 36.7 |
| Local | | - | 46.5 / 19.2 / 27.5 | 29.1 | 58.8 / 27.5 / 28.9 | 36.0 |

Table 2: Comparison of architectural tweaks to Local and GlobalLocal encoder. Staggering local blocks uses different blocks boundaries for different layers in block-local attention. Global information is incorporated in the decoder via an additional cross-attention before cross-attention over the encoded input.

In addition, we also introduce two simple variants of local attention Transformer encoders. First, we use a simple block-local Transformer (**Local**), where encoder input tokens are divided into non-overlapping blocks, and tokens can only attend to other tokens within the block. Second, we extend the local Transformer by adding a set of global tokens with learned embeddings, that can attend to and be attended from every encoder token (**Global-Local**). These components are similar to the sliding window attention and global token attention of Big Bird, ETC (Ainslie et al., 2020) and Longformer (Beltagy et al., 2020). However, we opt for the simpler block-local attention rather than sliding window attention, and compensate for the lack of overlapping blocks by staggering the local attention blocks, which we elaborate on in Section 3.2. As we show below, the performance is highly competitive despite its simplicity.

Results on short and long summarization tasks are shown in Table 1, with the relative training steps per second and memory consumed per device for fine-tuning on arXiv shown in the right-most columns. Among the short tasks, the full-attention Transformer performs best, followed by BigBird. On the long tasks, Big Bird and Global-Local models perform best, but Big Bird consumes significantly more memory and trains much more slowly than the other architectures. Conversely, although Performer has relatively low memory consumption and trains efficiently, it performs worst among the architectures tested by a noticeable margin.

On the other hand, Local and Global-Local encoders strike a good balance of performance and efficiency. The simple local attention encoder, which uses block-local attention, attains performance close to that of Big Bird while being much faster and using much less memory. Global-Local trades off a small amount of speed and memory for better performance, outperforming Big Bird.

**Takeaways:** Local attention is a strong baseline, and adding global tokens significantly improves performance. Both models are resource-efficient.

### 3.2 Local and Global-Local configurations

Given the good performance of both Local and Global-Local encoder variants, we next consider further architectural tweaks to these models.

First, we introduce *staggering* of local attention blocks. In block-local attention, tokens can only attend to other tokens within the same block. If the input tokens are divided up into the same blocks in every layer, this means that no information is exchanged across blocks through the entire encoder. To address this pitfall, we stagger attention blocks by shifting the block boundaries by half a block every other layer. We show an example of this in Figure 2. In practice, we implement this by padding

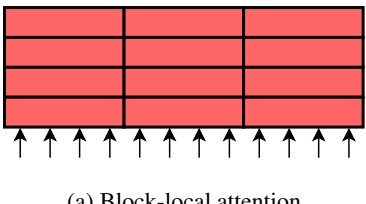

(a) Block-local attention

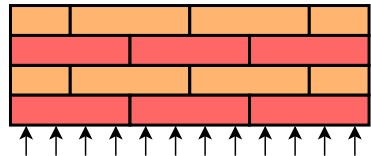

(b) Block-local attention with staggered blocks

Figure 2: In block-local attention (a), the same block boundaries are used across all layers, preventing information from being shared across blocks. Staggering the block boundaries (b) be shifting the boundaries every other layer allows for cross-block interactions with minimal additional computational cost or complexity.

the hidden representations on either side by half a block and masking accordingly.

Secondly, in the Global-Local model, the decoder only attends to the encoded token representations, and not the global token representations. We consider a variant where we supply the global token representations to the decoder and introduce a second cross-attention layer that attends only to the global tokens. Our goal is to allow the decoder to incorporate global information before performing cross-attention over the encoded sequence.

Results are shown in Table 2. We find that staggering local blocks noticably improves performance in both Local and Global-Local models. Performance improves even with Global-Local models, which already allow for cross-block interactions via global tokens, indicating that both model improvements are complementary. Conversely, incorporating global token information in the decoder did not lead to much performance improvement, particularly once staggered local blocks were used.

**Takeaways:** Staggering local attention blocks significantly improves performance, and is complementary to global tokens.

### 3.3 Global-Local: Block Size and Number of Global Tokens

Next, we vary the block size and number of global tokens for the Global-Local encoder, with results shown in Table 3.[2] Broadly, we find that increasing either block size or global tokens leads to improved performance, with a corresponding increase in memory consumption and computation time. However, the effect size from going to larger block sizes is not large, and saturates with larger block sizes or number of global tokens. As such, increasing either of these hyperparameters is ideal if

---

[2]Experiments with very small block sizes or number global tokens ran into memory issues, because TPUs pad small dimensions of arrays to certain minimum lengths, leading to larger than expected memory consumption.

resources allow, but is not a high priority compared to other model improvements. For the remainder of the ablation experiments, we use a block size of 64 and 32 global tokens for consistency.

**Takeaways:** Larger block sizes and/or number of global tokens leads to improved performance, although the effect saturates.

### 3.4 Other Architecture Modifications

We further investigate a of series architectural modifications to the encoder-decoder model, including the position encoding scheme (Table 8), scaling the encoder and decoder layers (Table 10) and using cross-attention in only a fraction of the decoder layers (Table 12). We find that the sinusoidal position encoding provide a good balance of performance and efficiency, and that a balanced encoder-decoder with full cross-attention generally performs the best. More details are provided in Appendix B.

### 3.5 Pretraining vs Fine-tuning Architectures

Previous works using efficient Transformer encoders have generally taken the parameters of a full-attention Transformer pretrained on a shorter sequences and adapted them to efficient architectures, either directly during fine-tuning (Zaheer et al., 2020) or with an intermediate stage of additional pretraining (Beltagy et al., 2020). In this section, we investigate if such an approach is optimal, or if models benefit from being pretrained with efficient encoders from the beginning. Note that we still perform pretraining on a short sequences (512 tokens), even with an efficient encoder.

We consider both pretraining with a Transformer and pretraining with the efficient architecture for both Local and Global-Local models. We also vary the block size, as the main difference between a Transformer and Local Transformer is the block size (aside from staggering, a Local model with block size 512 is equivalent to a dense

| Block Size | Global Tokens | arXiv | | GovReport | | Steps/s | Mem |
| :---: | :---: | :---: | :---: | :---: | :---: | :---: | :---: |
| | | R1 / R2 / RL | RG | R1 / R2 / RL | RG | | |
| 16 | 32 | 47.1 / 20.0 / 28.3 | 29.9 | 59.7 / 27.8 / 29.2 | 36.5 | 0.92 | 1.15 |
| | 64 | 46.8 / 19.7 / 28.0 | 29.6 | 60.8 / 28.6 / 30.0 | 37.4 | 0.75 | 1.54 |
| | 128 | 47.7 / 20.0 / 28.2 | 30.0 | 60.7 / 28.8 / 30.2 | 37.5 | 0.58 | 1.70 |
| 64 | 32 | 47.7 / 20.3 / 28.5 | 30.2 | 61.0 / 29.3 / 30.8 | 38.0 | 0.47 | 1.07 |
| | 64 | 47.4 / 20.2 / 28.5 | 30.1 | 60.9 / 29.1 / 30.7 | 37.9 | 0.94 | 1.10 |
| | 128 | **47.8 / 20.4 / 28.6** | **30.3** | 60.9 / 29.0 / 30.3 | 37.7 | 0.85 | 1.26 |
| 128 | 32 | 46.9 / 19.7 / 28.0 | 29.6 | 60.9 / 28.7 / 30.1 | 37.5 | **1.00** | **1.00** |
| | 64 | 47.4 / 20.2 / 28.4 | 30.1 | 60.9 / 28.9 / 30.8 | 37.8 | 0.96 | 1.05 |
| | 128 | 47.1 / 20.0 / 28.3 | 29.9 | 61.0 / 28.9 / 30.6 | 37.8 | 0.90 | 1.15 |
| 256 | 32 | 47.3 / 20.2 / 28.3 | 30.0 | 61.6 / 29.4 / 30.7 | 38.2 | 0.92 | 1.11 |
| | 64 | 47.2 / 20.2 / 28.4 | 30.0 | 59.2 / 28.6 / 30.5 | 37.2 | 0.88 | 1.16 |
| | 128 | 48.1 / 20.5 / 28.6 | 30.4 | **61.7 / 29.3 / 30.8** | **38.2** | 0.83 | 1.26 |

Table 3: Varying the block size and number of global tokens in Global-Local encoders. Training steps per second and memory are computed based on arXiv, and normalized to the run with Block Size=128 and Global Tokens=32.

Transformer), and hence the difference in block size also corresponds to the extent to which the model needs to adapt between architectures. When adapting from a pretrained Transformer encoder to a Global-Local architecture, because the Global-Local model relies on newly introduced global token embeddings, we initialize them by randomly sampling tokens from the vocabulary embeddings.

Results are shown in Table 11. For Local models, pretraining with local attention using small block sizes tends to hurt performance, but at moderate block sizes (e.g. 64) there is little difference between the two approaches. In contrast, for Global-Local pretraining with the efficient architecture tends to perform better. We hypothesize that this difference arises because of the learned global embedding tokens, which are randomly initialized when adapting from a pretrained Transformer and hence may benefit from pretraining and being jointly trained with the local attention.

**Takeaways:** For moderate block sizes, either pretraining or adapting to a Local encoder performs about equally well, but pretraining with a Global-Local encoder performs slightly better.

### 3.6 Pretraining Schemes

Up to this point, we have only considered pretraining with short sequences. We might expect that pretraining with longer sequences ought to improve performance on downstream long input summarization. However, pretraining only on long sequences is computationally expensive and requires a large collection of long input documents, which are relatively rarer. Long documents may also contain

different information from short documents, hence limiting training to only long inputs mae reduce the diversity of training data. Different long context Transformers have taken different approaches to pretraining on long inputs. For instance, Longformer (Beltagy et al., 2020) performed several additional stages of increasingly longer-sequence pretraining to adapt the initial RoBERTa to long sequence inputs. On the other hand, LongT5 (Guo et al., 2021) is pretrained exclusively with long input sequences. Others (Zaheer et al., 2020; Ivgi et al., 2022) perform no long input pretraining at all. In this section, we investigate how the balance of short and long pretraining impact downstream performance, and try to find the best trade-off between pretraining cost and downstream performance.

We consider two setups for pretraining: *short-input pretraining*, with 512 input tokens and 256 output tokens, and *long-input pretraining*, with 4096 input tokens and 256 output tokens. We describe the corresponding differences in data preprocessing in Section 2.1. We fix the number of input tokens seen during training, and vary configurations subject to this constraint. This constraint roughly proxies for the amount of compute consumed and corresponds to the number of input tokens seen during pretraining.[3]

We set our total input token budget at 131 billion tokens, which corresponds to 1 million steps with 512 input tokens, compared to the 500k steps in the above experiments. This larger budget ensures that when we only do long-input pretraining, the model

---

[3]If we instead fixed the number of training steps, long-input pretraining would consume far more compute for the same number of steps.

| Pretraining Scheme | Encoder | XSUM | | CNN/DM | | arXiv | | GovReport | |
|---|---|---|---|---|---|---|---|---|---|
| | | R1 / R2 / RL | RG | R1 / R2 / RL | RG | R1 / R2 / RL | RG | R1 / R2 / RL | RG |
| Short (50%) | Local | 38.4 / 15.8 / 30.6 | 26.5 | 39.2 / 18.1 / 27.9 | 27.1 | 46.8 / 19.7 / 28.0 | 29.6 | 60.1 / 28.3 / 29.8 | 37.0 |
| | Global-Local | 39.4 / 16.5 / 31.5 | 27.4 | 39.1 / 18.6 / 28.3 | 27.4 | 47.7 / 20.4 / 28.6 | 30.3 | 61.9 / 29.6 / 30.8 | 38.4 |
| Short (100%) | Local | 39.2 / 16.3 / 31.3 | 27.1 | 39.2 / 18.6 / 28.3 | 27.4 | 46.9 / 19.7 / 28.0 | 29.6 | 60.1 / 28.3 / 29.8 | 37.0 |
| | Global-Local | **39.9 / 17.0 / 31.9** | **27.9** | 39.8 / 18.6 / 28.3 | 27.6 | 48.1 / 20.5 / 28.7 | 30.5 | 61.9 / 29.6 / 30.8 | 38.4 |
| Short (75%) → Long (25%) | Local | 38.8 / 15.9 / 30.7 | 26.7 | 39.1 / 18.2 / 28.0 | 27.1 | 47.5 / 20.1 / 28.2 | 30.0 | 60.6 / 28.9 / 30.6 | 37.7 |
| | Global-Local | 39.6 / 16.8 / 31.7 | 27.6 | **39.8 / 18.8 / 28.5** | **27.7** | 48.4 / 20.7 / 28.8 | 30.7 | 61.8 / 29.8 / 31.1 | 38.5 |
| Short (50%) → Long (50%) | Local | 38.4 / 15.7 / 30.5 | 26.4 | 39.4 / 18.1 / 27.9 | 27.1 | 47.7 / 20.2 / 28.3 | 30.1 | 60.9 / 29.1 / 30.7 | 37.9 |
| | Global-Local | 39.3 / 16.4 / 31.4 | 27.3 | 39.4 / 18.3 / 28.1 | 27.3 | **48.4 / 20.9 / 29.1** | **30.9** | **61.7 / 30.0 / 31.2** | **38.7** |
| Long (100%) | Local | 36.0 / 14.0 / 28.6 | 24.3 | 38.4 / 17.7 / 27.4 | 26.5 | 46.7 / 19.5 / 27.7 | 29.3 | 59.8 / 28.0 / 29.5 | 36.7 |
| | Global-Local | 36.4 / 14.3 / 28.9 | 24.7 | 38.5 / 17.8 / 27.5 | 26.6 | 47.3 / 19.9 / 28.1 | 29.8 | 61.1 / 29.1 / 30.7 | 37.9 |

Table 4: Comparison of different pretraining formats, given a input token budget of 131B tokens, which corresponds to 1M steps with 512 input tokens. Short pretraining uses 512 input tokens, whereas long pretraining uses 4096 input tokens.

is still pretrained for a reasonable number of steps. We consider four pretraining configurations:

- Short-input for 100% of tokens (1M steps)

- Short-input for 75% of tokens (98.3B, 750k steps), then long-input for 25% of tokens (32.8B, 31.25k steps)

- Short-input for 50% of tokens (62.5B, 500k steps), then long-input for 50% of tokens (62.5B, 62.5k steps)

- Long-input for 100% of tokens (125k steps)

We compare the performance of the different pretraining scehemes in Table 4. We also include short-input pretraining for 500k steps for comparison. First, comparing short-input pretraining for 500k and 1M steps, we find that more pretraining still improves performance, indicating that our base models may still be undertrained at 500k steps. Second, long-input pretraining performs consistently worse than the other variants, which we attribute having fewer training steps, again highlighting the issue of potential undertraining. For the middle three configurations, on the long tasks, all three non-long-only variants atttain similar scores, with more long-input pretraining having slightly better performance, particularly on the ROUGE-2 and ROUGE-L scores. While the small absolute differences in scores make it hard to draw strong conclusions, we lean towards the conclusion that adding a short phase of long input pretraining can improve performance on long input summarization tasks.

**Takeaways:** Given a fixed compute budget, allocating some training steps to long-input training can improve performance, although the optimal allocation is difficult to determine. Exclusively long pretraining results in worse performance.

## 4  PEGASUS-X

Based on our findings, we settle on the following recipe for adapting PEGASUS models (Zhang et al., 2020) to long sequence summarization.

- We use a Global-Local architecture with block staggering, a large number of global tokens, and large block sizes during pretraining.

- We perform additional long input pretraining on 4096 token inputs for 300k steps.

- We extend input sequences up to 16384 input tokens in fine-tuning, depending on the task.

We experiment with two model sizes: **PEGASUS-X** (PEGASUS e**X**tended) based on PEGASUS$_{Large}$, and PEGASUS-X$_{Base}$ based on a newly trained PEGASUS$_{Base}$ model which we call PEGASUS$_{Base+}$.[4]

We initialize the weights of PEGASUS-X and PEGASUS-X$_{Base}$ with the pretrained weights of PEGASUS$_{Large}$ and PEGASUS$_{Base+}$ respectively. Only two new sets of parameters are introduced: global token embeddings, and a new LayerNorm for the global input representations in each Transformer layer. This is ∼1M more parameters for PEGASUS-X$_{Base}$ and 2M more for PEGASUS-X. We initialize the global token embeddings by randomly sampling tokens from the input embeddings, and we initialize the LayerNorm weights with the regular input LayerNorm weights.

The task- and model-specific hyperparameters for fine-tuning can be found in Appendix 15. For this section, we report ROUGE-Lsum[5] rather than ROUGE-L for consistency with the metrics reported in other papers and leaderboards.

---

[4]See Appendix C.
[5]https://github.com/google-research/

|  | PEGASUS-X$_{Base}$ | PEGASUS-X |
|---|---|---|
| # Parameters | 272M | 568M |
| # Global Tokens | 128 | 128 |
| Block Size | 512 | 512 |
| Batch Size | 512 | 1024 |
| Additional Pretraining | 300K steps | 300K steps |

Table 5: Hyperparameters of PEGASUS-X Models

## 4.1 Results on Summarization Tasks

**Long summarization tasks** In Table 6, we compare the performance of PEGASUS models to those of PEGASUS-X on three long-input summarization tasks: arXiv, Big Patent and PubMed. In all three tasks, we see significant improvements in performance of PEGASUS-X$_{Base}$ over PEGASUS$_{Base+}$, and PEGASUS-X over PEGASUS$_{Large}$. To isolate the impact of additional long input pretraining compared to only switching the architecture during fine-tuning, we also include evaluation on the PEGASUS models using the Global-Local architecture with no further pretraining, which we list in the table as PEGASUS$_{Base+}$ + Global-Local.

We also compare to reported results of Big Bird-PEGASUS[6] (Zaheer et al., 2020), LED (Beltagy et al., 2020), Top-Down Transformer (Pang et al., 2022) with both Average-Pool (AvgP) and Adaptive-Pool (AdaP) variants, BART-LS (Xiong et al., 2022a), LongT5-Large and XL, and SLED (Ivgi et al., 2022). LED, Top-Down and SLED are initialized with BART$_{Large}$ weights with no additional pretraining on long input sequences. BART-LS is concurrent work that also incorporates staggered block-local attention and addition long-sequence pretraining, in addition to pooling layers and different pretraining data.

PEGASUS-X outperforms Big Bird-PEGASUS on all tasks, and Top-Down-AvgP on both compared tasks. Although Top-Down-AdaP outperforms PEGASUS-X, it uses a much more complex fine-tuning setup, using an importance tagger on reference summaries to construct token pooling weights, whereas PEGASUS-X only uses standard fine-tuning. Even so, PEGASUS-X still outperforms Top-Down-AdaP on PubMed. PEGASUS-X outperforms BART-LS on PubMed and slightly underperforms on arXiv; as mentioned above,

PEGASUS-X and BART-LS share many similarities, and we see the strong performance of BART-LS as confirmation of the efficacy of parts of our recipe for longer sequence models. PEGASUS-X also outperforms LongT5 on both arXiv and PubMed, despite both compared LongT5 models having more parameters. However, we find that LongT5 performs much better on BigPatent, which is a largely extractive summarization task. We hypothesize that a larger hidden size may improve extraction over very long sequences.

**Short summarization tasks** We show in Table 14 the performance of PEGASUS and PEGASUS-X models on shorter summarization tasks, where there is a slight regression in performance of both PEGASUS-X models compared to their PEGASUS equivalents. We hypothesize that long input pretraining might negatively impact the performance on shorter input tasks because of the data filtering for long documents, resulting in a potentially less diverse training data distribution.

## 4.2 SCROLLS Summarization Tasks

We report the performance of the PEGASUS-X models on the summarization tasks in the recently introduced SCROLLS benchmark in Table 7. This includes GovReport (Huang et al., 2021), the ForeverDreaming subset of SummScreen (Chen et al., 2022), and QMSum (Zhong et al., 2021).

PEGASUS-X outperforms all other models on GovReport, setting the state of the art on the dataset.[7] It also performs comparably to both LongT5$_{Large}$ and Top-Down-AvgP on SummScreen/FD, although it underperforms LongT5 models and BART-LS on QMSum. Moreover, PEGASUS-X$_{Base}$ also performs competitively, outperforming both LongT5 models on GovReport, and only a small margin behind PEGASUS-X on all three tasks. PEGASUS-X$_{Base}$ also outperforms BART$_{Large}$-SLED, a larger model with a similar 16K input length.

## 5 Pertinent Related Work

Many works such as Zaheer et al. (2020), Beltagy et al. (2020), Ivgi et al. (2022) have investigated extending short input models to longer sequences using efficient attention mechanisms. In closely comparable work, Guo et al. (2021) pretrained a T5 model on long sequences from scratch, incor-

---

google-research/blob/master/rouge/README.md#two-flavors-of-rouge-l

[6]Big Bird-PEGASUS only has a context of 3072 tokens, likely due to the larger memory consumption of Big Bird.

[7]As of 08/08/2022

| Model | #Params | arXiv R1 / R2 / RLs | RG | Big Patent R1 / R2 / RLs | RG | PubMed R1 / R2 / RLs | RG |
|---|---|---|---|---|---|---|---|
| PEGASUS$_{Base}$ | 271M | 34.8 / 10.2 / 22.5* | 20.0* | 43.5 / 20.4 / 31.8* | 30.5* | 40.0 / 15.2 / 25.2* | 24.8* |
| PEGASUS$_{Base+}$ | 271M | 42.2 / 15.8 / 37.3 | 29.2 | 51.2 / 32.6 / 41.0 | 40.9 | 44.1 / 18.3 / 40.1 | 31.9 |
| PEGASUS$_{Base+}$ + Global-Local | 272M | 47.6 / 20.2 / 42.4 | 34.4 | 58.1 / 39.5 / 47.2 | 47.7 | 47.3 / 21.4 / 43.0 | 35.2 |
| PEGASUS-X$_{Base}$ | 272M | 49.4 / 21.6 / 44.0 | 36.1 | 61.3 / 42.6 / 50.1 | 50.8 | 49.6 / 23.6 / 45.2 | 37.5 |
| PEGASUS$_{Large}$ | 567M | 44.7 / 17.2 / 25.7* | 27.0* | 53.4 / 32.9 / 42.1* | 42.0* | 45.1 / 19.6 / 27.4* | 28.9* |
| PEGASUS-X | 568M | 50.0 / 21.8 / 44.6 | 36.5 | 64.8 / 47.5 / 54.3 | 55.1 | **51.0 / 24.7 / 46.6** | **38.9** |
| BART-LS | 460M | 50.2 / 22.1 / 45.4 | 36.9 | –.- / –.- / –.- | –.- | 50.3 / 24.3 / 46.3 | 38.4 |
| Longformer Encoder-Decoder | 464M | 46.6 / 19.6 / 41.8 | 33.7 | –.- / –.- / –.- | –.- | –.- / –.- / –.- | –.- |
| Top-Down (AvgP) | 464M | 48.7 / 20.7 / 43.9 | 35.4 | –.- / –.- / –.- | –.- | 48.3 / 21.4 / 44.2 | 35.7 |
| Top-Down (AdaP) | 464M | **51.0 / 21.9 / 45.6** | **37.1** | –.- / –.- / –.- | –.- | 51.1 / 23.3 / 46.5 | 38.1 |
| Big Bird-Pegasus | 567M | 46.6 / 19.0 / 41.8 | 33.3 | 60.6 / 42.5 / 50.1 | 50.5 | 46.3 / 20.7 / 42.3 | 34.4 |
| LongT5$_{Large}$ | 770M | 48.3 / 21.6 / 44.1 | 35.8 | 70.4 / 56.8 / 62.7 | 63.1 | 50.0 / 24.7 / 46.5 | 38.6 |
| LongT5$_{XL}$ | 3B | 48.4 / 21.9 / 44.3 | 36.1 | **76.9 / 66.1 / 70.8** | **71.1** | 50.2 / 24.8 / 46.7 | 38.7 |

Table 6: Comparison on long summarization tasks (Test sets). Results for other models are taken from their respective papers. *: PEGASUS (Zhang et al., 2020) only reports ROUGE-L and not ROUGE-LSum.

| Model | #Params | GovReport R1 / R2 / RL | RG | SummScreen/FD R1 / R2 / RL | RG | QMSum R1 / R2 / RL | RG |
|---|---|---|---|---|---|---|---|
| PEGASUS-X$_{Base}$ | 272M | 59.3 / 29.3 / 30.9 | 37.7 | 35.0 / 8.9 / 20.4 | 18.5 | 32.9 / 9.8 / 21.4 | 19.0 |
| PEGASUS-X | 568M | **60.3 / 30.0 / 31.5** | **38.5** | 35.7 / 9.1 / 20.6 | 18.8 | 33.2 / 9.6 / 21.6 | 19.0 |
| BART$_{Large}$-SLED | 406M | 58.0 / 26.9 / 27.6 | 35.1 | 33.8 / 8.0 / 18.5 | 17.1 | 32.1 / 10.2 / 21.0 | 19.0 |
| BART-LS | 460M | 59.4 / 29.8 / 30.8 | 37.9 | **37.7 / 10.2 / 21.5** | **20.2** | 35.1 / 12.0 / 23.3 | 21.4 |
| Top-Down-AvgP | 464M | –.- / –.- / –.- | –.- | 35.8 / 8.9 / 30.6* | 21.4* | –.- / –.- / –.- | –.- |
| Top-Down-AdaP | 464M | –.- / –.- / –.- | –.- | 36.8 / 9.2 / 31.1* | 21.9* | –.- / –.- / –.- | –.- |
| LongT5$_{Large}$ | 770M | 54.2 / 27.8 / 29.8 | 35.5 | 35.6 / 9.2 / 21.2 | 19.1 | **35.1 / 12.0 / 23.3** | **21.4** |
| LongT5$_{XL}$ | 3B | 54.7 / 28.2 / 30.2 | 36.0 | 35.8 / 9.6 / 21.1 | 19.4 | 34.9 / 11.8 / 23.5 | 21.3 |
| UL2 | 20B | 53.6 / 26.1 / 28.8 | 34.3 | 32.9 / 7.8 / 19.4 | 17.1 | 31.1 / 8.5 / 20.4 | 17.5 |

Table 7: Comparison on SCROLLS benchmark (Summarization tasks, Test sets). Results for SLED, BART-LS, LongT5 and UL2 models are taken from the SCROLLS benchmark leaderboard. *: Top-Down (Pang et al., 2022) reports much higher scores for ROUGE-L on SummScreen/FD than any other model, and may have been computed with a variant of ROUGE-L that involves splitting on sentences rather than newlines.

porating sliding window attention and global representations. However, pretraining only on long sequences significantly increases the pretraining time, and as we show in Section 3.6, pretraining first on short inputs and then subsequently on long inputs is much more cost efficient.

In concurrent work released shortly before this submission deadline, Xiong et al. (2022a) also investigated extending short input Transformer models for long input tasks. While they focus on BART rather than PEGASUS, they similarly find that global tokens, staggered block-local attention, and extended pretraining greatly improve performance, lending further support to our findings. Their final model also incorporates pooling layers and is trained on different data.

A broader treatment of related work can be found in Appendix A.

# 6 Conclusion

In this work, we investigate a range of proposed improvements to Transformer models to effectively and economically handle long inputs in summarization tasks. Through extensive ablation experiments, we find a simple but effective recipe for extending short-input models to tackle long-input summarization. Based on our findings, we introduce PEGASUS-X, an extended version of PEGASUS with a modified architecture and additional long-sequence pretraining. We show that PEGASUS-X sets the state of the art on two long input summarization tasks (GovReport and PubMed) and performs competitively on many others, even despite being much smaller than some compared models. Our findings can be extended to models in other domains beyond summarization, both for pretraining long input models from scratch as well as extending already pretrained short sequence models.

## Limitations

### Challenges of Evaluating Long-Document Summarization Models

One limitation of our work is that evaluation of long-document summarization models is challenging, and while we evaluate on the widely used benchmarks for long-document summarization models, we highlight here the difficulties of measuring the capabilities of such models. In addition to the widely accepted issues with automatic evaluation of model-generated summaries with metrics such as ROUGE, long-document summarization brings about new challenges. In particular, there are relatively fewer long-document summarization tasks available to evaluate models on, and many of them (e.g. arXiv, Pub Med, SummScreen) are constructed by repurposing existing data and proxies for summaries (e.g. abstracts) rather than explicitly written summaries. As such, the available datasets for summarization reflect the data that is easy to repurpose into summarization rather than practical downstream summarization settings; in other words, the available evaluation datasets may not match the distribution of data or settings where such models are realistically used.

On scoring generations, human evaluation should ideally be conducted to measure the quality of model-generated summaries. However, the much longer input texts also means that human evaluation of summaries becomes much more expensive and onerous, as raters would need to read the whole input before judging the quality of the summary.

More discussion on the challenges of evaluating long-document summarization models can be found in Wang et al. (2022).

### Findings May Not Generalize to Other Tasks

We have confined our study to summarization tasks, as it matches our goal of investigating the ability for models to process large input contexts, with less focus on generating long outputs. We acknowledge that our ablation studies and experiments are focused solely on summarization tasks, and that our findings may not directly apply or extend to other long-input language tasks.

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

## A  Full Related Work

**Long Document Summarization**  Several new long input summarization datasets and benchmarks have been recently introduced, providing better measures of long input summarization capability as well as prompting new interest in this research direction. The BookSum dataset (Kryściński et al., 2021) consists of paragraph, chapter, and full summaries of books on Project Gutenberg based on web-scraped educational website. (Chen et al., 2022) consists of television show transcripts and episode summaries based on web-scraped fan-written summaries. The SCROLLS benchmark (Shaham et al., 2022) and the MuLD benchmark (Hudson and Al Moubayed, 2022) consist of multiple natural language tasks with long inputs, including long input summarization. The SQuALITY dataset (Wang et al., 2022) consists of question-focused summaries of Project Gutenberg stories, where annotators write summaries based on different questions that cover different aspects of the same story.

**Efficient Transformers**  Many efficient Transformer variants have been introduced in recent years (Tay et al., 2020), and we discuss here the works more relevant to this manuscript. (Beltagy et al., 2020) use global tokens as well as a sliding window local attention, implemented using custom CUDA kernels. The ETC model (Ainslie et al., 2020) uses both global tokens and block-wise sliding window local attention, although the global attention is incorporated based on the first few tokens of a sequence, rather than separately learned global tokens. Zaheer et al. (2020) extend ETC by adding random attention blocks, but we found that this significantly increases code complexity and computational cost. Guo et al. (2021) similarly extend ETC's block-wise sliding window attention, but computes transient "global token" representations by pooling over blocks of tokens. Pang et al. (2022) propose to augment the Longformer encoder-decoder with additional pooling layers to improve long-sequence summarization performance. Ivgi et al. (2022) propose an alternative approach to sparse attention via encoding overlapping chunks and fusing information across chunks int he decoder. We highlight that while the final Global-Local model architecture that we settle on shares similarity with several other proposed efficient Transformer architectures, our key con-

tribution lies in our extensive ablation study that identifies architectural tweaks that improve and, just as importantly, do not improve downstream performance.

Among the listed model architectures for long input summarization, LongT5 (Guo et al., 2021) is the most similar to PEGASUS-X, sharing a similar encoder-decoder architecture, a similar training objective in generating masked sentences, and a mix of local attention and global information sharing for the encoder. We briefly highlight the key differences between the two models. Firstly, LongT5 trains from scratch on long sequences, whereas we initialize our model weights with PEGASUS weights (which is trained on short sequences) before doing additional pretraining on long input sequences. This significantly reduces the overall pretraining cost, as short sequence pretraining and be performed much more economically. LongT5 also uses the T5 relative position biases whereas PEGASUS-X uses sinusoidal position embeddings–as shown in Section B.1, T5 relative position biases perform slightly better but are significantly slower. The efficient encoder architecture between the two models is also different: LongT5 uses a transient global representations based on pooling chunks of tokens, whereas PEGASUS-X uses learned global token embeddings. LongT5 also uses a sliding window local attention based on ETC (Ainslie et al., 2020), whereas we use a simpler block-local attention with staggered blocks. Lastly, the largest LongT5 model is 3B parameters, more than $5\times$ the size of PEGASUS-X.

More broadly, Tay et al. (2021) compare a variety of efficient Transformer architectures on a set of tasks designed to probe long-sequence processing capability, evaluating the different models on both performance as well as computation requirements. Tay et al. (2022) further evaluate the scaling properties of novel Transformer architectures, finding that deviating from full attention tends to hurt downstream performance. Xiong et al. (2022b) showed that simple local attention variants can be highly competitive with more complex sparse attention schemes, consistent with our findings.

## B  Details of Architecture Modification Experiments

### B.1  Position Encoding Schemes

New position encoding schemes encoding schemes such as RoPE (Su et al., 2021) and ALiBi (Press

| Position Encoding | XSUM | | CNN/DM | | arXiv | | GovReport | | |
|---|---|---|---|---|---|---|---|---|---|
| | R1 / R2 / RL | RG | R1 / R2 / RL | RG | R1 / R2 / RL | RG | R1 / R2 / RL | RG | Step/s |
| None | 34.3 / 12.5 / 26.8 | 22.6 | 25.6 / 7.8 / 17.7 | 15.2 | 36.1 / 9.8 / 22.0 | 19.8 | 38.3 / 13.2 / 18.7 | 21.1 | 0.96 |
| Sinusoidal | 39.8 / 16.9 / 31.8 | 27.8 | **40.0 / 18.6 / 28.4** | **27.6** | 44.5 / 17.6 / 26.7 | 27.6 | 40.0 / 18.8 / 22.3 | 25.6 | 0.96 |
| T5 | **40.1 / 17.1 / 32.0** | **28.0** | 39.8 / 18.8 / 28.6 | 27.8 | **44.9 / 17.9 / 26.8** | **27.8** | **40.2 / 19.5 / 22.9** | **26.2** | 0.53 |
| RoPE | 39.8 / 16.9 / 31.8 | 27.8 | 39.2 / 18.7 / 28.5 | 27.5 | 43.5 / 17.2 / 26.5 | 27.1 | 40.0 / 19.1 / 22.6 | 25.8 | 0.85 |
| Absolute | 39.1 / 16.4 / 31.3 | 27.2 | 39.7 / 18.7 / 28.5 | 27.7 | 44.3 / 17.5 / 26.5 | 27.4 | 38.6 / 17.5 / 21.1 | 24.2 | **1.00** |

Table 8: Comparison of position encodings schemes for a Transformer encoder-decoder. Training steps per second are computed based on arXiv summarization. Absolute position embeddings are replicated to longer input sequences, following Beltagy et al. (2020). Training steps per second is computed based on arXiv, and normalized to the run with absolute position embeddings.

| Position Encoding | arXiv | | GovReport | |
|---|---|---|---|---|
| | R1 / R2 / RL | RG | R1 / R2 / RL | RG |
| Factor=10000 | 48.1 / 20.4 / 28.6 | 30.4 | 60.9 / 29.3 / 30.8 | 38.0 |
| Factor=50000 | 48.1 / 20.4 / 28.6 | 30.4 | 61.4 / 29.5 / 30.9 | 38.3 |

Table 9: Comparison of different scaling constants in sinusoidal position encodings.

et al., 2022) have garnered recent attention, showing improved performance on downstream evaluations. As input sequence lengths have gotten much longer, and in particular longer than the dimensions of hidden representations, previous choices of position encoding may no longer be optimal. Moreover, relative position encodings such as RoPE, T5 and ALiBi may be better suited for adapting models to different input lengths between pretraining and fine-tuning. Hence, this is a good opportunity to revisit the choice of positioning encoding schemes in encoder models.

Because of the more complex interaction between local attention blocks and relative position encoding implementations, we conduct a preliminary investigation with a full-attention Transformer. We pretrain with an input length of 512, and fine-tune with an input length of 2048 for the long sequence tasks – this experiment also tests the propensity for position encodings to be adapted to longer sequences downstream. In addition to the sinusoidal position encoding used in PEGA-SUS and Vaswani et al. (2017), we also consider the bucket-based relative position encoding scheme of T5, RoPE, absolute position embeddings, and no position encoding as a baseline. For absolute position embeddings, we follow the recipe of Beltagy et al. (2020) and duplicate the learned position embeddings to handle longer sequences before fine-tuning. The chosen position encoding scheme is applied to all parts of the model, including both the encoder and the decoder. We do not experiment with ALiBi, as we found no natural way to adapt

ALiBi to cross-attention.

Our results are shown in Table 8. We find that although T5 performs the best, it is also almost twice as slow as the other position encoding schemes, which is consistent with the findings of Press et al. (2022). Sinusoidal position encodings and RoPE perform only slightly worse than T5 with much better efficiency, making them more desirable choices. Given the much simpler implementation of sinusoidal position encodings, we opt to stick with them for the remainder of the experiments.

**Takeaways:** Sinusoidal position encodings still remain a good choice for long input Transformers.

### B.2 Scaling Encoder and Decoder Layers

Scaling laws (Kaplan et al., 2020; Ghorbani et al., 2021; Zhang et al., 2022) that describe the empirical relationship between model sizes and performance have proven surprisingly consistent and gotten significant attention in recent years. We present in this section a small set of scaling experiments, exploring the distribution of layers between encoder and decoder.

Our results are shown in Table 10. In the top half, we fix the total number of layers to 24, and consider both encoder-heavy and decoder-heavy distributions, for both Local and Global-Local models. We observe that impact of distribution of encoder and decoder layers on performance is relatively small. For Local models, we see a slight boost from decoder-heavy models. For Global-Local models, we observe that a balanced encoder-decoder outperforms encoder- and decoder-heavy models, both of

| Architecture | Enc | Dec | XSUM | | CNN/DM | | arXiv | | GovReport | |
|---|---|---|---|---|---|---|---|---|---|---|
| | | | R1 / R2 / RL | RG | R1 / R2 / RL | RG | R1 / R2 / RL | RG | R1 / R2 / RL | RG |
| Local | 18 | 6 | 37.4 / 15.0 / 29.7 | 25.5 | 39.0 / 18.2 / 27.9 | 27.0 | 46.0 / 19.4 / 27.6 | 29.1 | 58.9 / 27.4 / 29.1 | 36.1 |
| | 12 | 12 | 37.5 / 14.9 / 29.7 | 25.5 | 38.5 / 18.0 / 27.6 | 26.7 | 45.4 / 18.9 / 27.3 | 28.6 | 59.2 / 27.6 / 29.3 | 36.3 |
| | 6 | 18 | 37.7 / 15.1 / 29.9 | 25.7 | 38.5 / 18.1 / 27.7 | 26.9 | 46.3 / 19.3 / 27.6 | 29.1 | 59.4 / 27.8 / 29.5 | 36.5 |
| Global-Local | 18 | 6 | **38.6 / 15.9 / 30.9** | **26.7** | 39.2 / 18.5 / 28.2 | 27.3 | 47.3 / 20.1 / 28.3 | 30.0 | 60.2 / 28.7 / 30.6 | 37.5 |
| | 12 | 12 | 38.6 / 15.9 / 30.7 | 26.6 | **40.0 / 18.6 / 28.3** | **27.6** | 47.5 / 20.1 / 28.3 | 30.0 | **61.1 / 29.3 / 30.7** | **38.1** |
| | 6 | 18 | 37.7 / 15.1 / 29.9 | 25.7 | 38.5 / 18.1 / 27.7 | 26.9 | 46.4 / 19.5 / 27.9 | 29.3 | 60.3 / 28.6 / 30.0 | 37.2 |
| Global-Local | 18 | 12 | 38.5 / 15.7 / 30.6 | 26.4 | 38.7 / 18.4 / 28.1 | 27.1 | 47.3 / 20.0 / 28.3 | 29.9 | 60.2 / 29.2 / 31.0 | 37.9 |
| | 12 | 18 | 38.6 / 15.8 / 30.5 | 26.5 | 38.6 / 18.3 / 28.0 | 27.0 | **47.5 / 20.3 / 28.5** | **30.2** | 60.9 / 29.0 / 30.4 | 37.7 |

Table 10: Varying the distribution of encoder/decoder layers)

which perform about comparably.

We also consider cases where we further increase the size of either the encoder or decoder to 18 layers, shown in the second half of Table 10. We observe no improvement in performance over the 12/12-layer encoder-decoder, and suspect that other hyperparameters (e.g. hidden size) might be the bottleneck rather than the number of layers.

We highlight here that because of the asymmetry of the input and output lengths, there are different computational trade-offs to different balances of encoder and decoder layers. Encoder-heavy models require more memory because of the long input sequences, whereas decoder-heavy models are relatively slower at inference because of the autoregressive nature of decoding. Given the relatively small difference in the margin of performance, memory or computational constraints may outweigh the performance differences in practical scenarios.

**Takeaways:** A balanced encoder-decoder performs best, but the difference in performance may be outweighed by other resource considerations.

### B.3 Partial Cross Attention

Given the use of an efficient attention architecture, which has memory consumption scale linearly rather than quadratically in input sequence length, another major memory bottleneck is the encoder-decoder cross-attention. Because each decoder layer attends separately to the long encoder representations, and the attention is dense, this is a large contiguous chunk of memory that we could seek to reduce.

Perceiver AR (Hawthorne et al., 2022) demonstrated strong performance by using only a single cross-attention at the bottom layer of an autoregressive language model. Based on these results, we investigate the impact of only having cross-attention on a subset of decoder layers. In Table 12, we show the results of pretraining and fine-tuning Global-Local models with cross-attention only on specific layers on a variety of configurations. We find that reducing the number of cross-attention layers leads to a drop in performance, but the impact on performance is smaller than expected. For instance, with only cross-attention on the first and sixth layer, the Global-Local model still outperforms a Local model. The reduction of cross-attention layers also leads to a corresponding improvement in training step and reduction in memory consumption.

Given the small drop in performance from using fewer decoder layers with cross-attention, we consider the viability of dropping cross-attention layers after pretraining. In other words, we take a Global-Local model pretrained with full cross-attention, drop the cross-attention for a subset of layers, and fine-tune directly. Our results are shown in Table 13. We find that dropping the cross-attention after pretraining again only leads to a small (additional) dip in performance. This indicates that dropping cross-attention may be a viable strategy for further reducing memory requirements for an existing pretrained model with a small performance trade-off, and pretraining a separate model from scratch is not necessary.

**Takeaways:** Dropping cross-attention for a fraction of decoder layers can reduce memory consumption at the cost of slight performance regression. Cross-attention can be dropped after pretraining, with an associated performance trade-off.

### B.4 Comparison on short summarization tasks

### C PEGASUS$_{\text{Base+}}$

In a similar finding as Hoffmann et al. (2022), we found that PEGASUS$_{\text{Base}}$ benefits from training on significantly more tokens. As such, we trained a PEGASUS$_{\text{Base}}$ for a much larger number of to-

| Pretraining → Fine-tuning | Block Size | arXiv | | GovReport | |
|---|---|---|---|---|---|
| | | R1 / R2 / RL | RG | R1 / R2 / RL | RG |
| Transformer → Local | 16 | 46.4 / 19.6 / 27.9 | 29.4 | 59.6 / 28.2 / 29.9 | 36.9 |
| | 64 | 46.5 / 19.5 / 27.8 | 29.3 | 59.5 / 28.0 / 29.6 | 36.7 |
| | 256 | 46.8 / 19.7 / 28.0 | 29.6 | 59.8 / 28.0 / 29.8 | 36.8 |
| Local → Local | 16 | 45.9 / 19.1 / 27.5 | 28.9 | 59.0 / 27.5 / 29.3 | 36.2 |
| | 64 | 46.5 / 19.5 / 27.8 | 29.3 | 59.7 / 28.1 / 29.8 | 36.8 |
| | 256 | 47.1 / 19.9 / 28.1 | 29.8 | 59.7 / 28.5 / 30.3 | 37.2 |
| Transformer → Global-Local | 16 | 46.0 / 19.2 / 27.5 | 29.0 | 60.3 / 28.2 / 29.8 | 37.0 |
| | 64 | 47.0 / 20.0 / 28.2 | 29.8 | 60.8 / 28.7 / 30.1 | 37.4 |
| | 256 | 47.6 / 20.3 / 28.4 | 30.2 | 60.8 / 28.7 / 30.0 | 37.4 |
| Global-Local → Global-Local | 16 | 47.1 / 20.0 / 28.3 | 29.9 | 59.7 / 27.8 / 29.2 | 36.5 |
| | 64 | **47.7 / 20.3 / 28.5** | **30.2** | 61.0 / 29.3 / 30.8 | 38.0 |
| | 256 | 47.3 / 20.2 / 28.3 | 30.0 | **61.6 / 29.4 / 30.7** | **38.2** |

Table 11: Comparison of adapting models architectures between pretraining and fine-tuning.

| Cross-Attention | XSUM | | CNN/DM | | arXiv | | GovReport | | Step/s | Mem |
|---|---|---|---|---|---|---|---|---|---|---|
| | R1 / R2 / RL | RG | R1 / R2 / RL | RG | R1 / R2 / RL | RG | R1 / R2 / RL | RG | | |
| Full | **38.8 / 16.0 / 31.0** | **26.8** | 39.5 / 18.6 / 28.4 | 27.5 | 47.7 / 20.4 / 28.6 | 30.3 | **61.3 / 29.4 / 30.8** | **38.1** | 1.00 | 1.00 |
| Cross[0,2,4,6,8,10] | 38.3 / 15.6 / 30.5 | 26.3 | **39.8 / 18.8 / 28.5** | **27.7** | **48.1 / 20.4 / 28.6** | **30.4** | 61.0 / 29.0 / 30.7 | 37.9 | 1.10 | 0.90 |
| Cross[0,3,6,9,11] | 38.0 / 15.3 / 30.2 | 26.0 | 38.8 / 18.4 / 28.1 | 27.2 | 46.9 / 19.9 / 28.2 | 29.7 | 60.1 / 28.6 / 30.2 | 37.3 | 1.15 | 0.88 |
| Cross[0,4,8,11] | 37.8 / 15.3 / 30.1 | 25.9 | 38.5 / 18.1 / 27.9 | 26.9 | 47.6 / 20.2 / 28.4 | 30.1 | 60.9 / 28.9 / 30.3 | 37.6 | 1.15 | 0.86 |
| Cross[0,6,11] | 37.4 / 14.8 / 29.7 | 25.4 | 38.8 / 18.1 / 27.9 | 27.0 | 46.9 / 19.7 / 28.1 | 29.6 | 60.3 / 28.5 / 30.2 | 37.3 | 1.18 | 0.87 |
| Cross[0,6] | 37.5 / 14.9 / 29.7 | 25.5 | 38.3 / 18.0 / 27.8 | 26.8 | 47.1 / 19.8 / 28.1 | 29.7 | 60.4 / 28.1 / 29.7 | 36.9 | **1.21** | **0.85** |

Table 12: Comparison of models with cross-attention only in a subset of the 12 decoder layers. Training steps per second and memory are computed based on arXiv, and normalized to the Cross[0,6] run.

kens (the same as PEGASUS$_{Large}$), which achieves much better performance than the previously released PEGASUS$_{Base}$ model.

## D  Encoder Architecture Hyperparameters

For experiments in Section 3.1, BigBird, Local and Global-Local all use a block size of 64. BigBird and Global-Local also use 32 global tokens. Performer uses 256 random features.

## E  Fine-tuning Hyperparameters

For arXiv, we fine-tune with an input length of up to 16384 tokens and 256 output tokens, while for GovReport we use an input length of 10240 input tokens and 1024 output tokens given the longer summaries for the task. For XSUM and CNN/Daily Mail, with use an input length of 512, and output lengths of 64 and 128 respectively, following PEGASUS hyperparameters. The full set of hyperparameters for fine-tuning models are shown in Table 15.

## F  Engineering Details

The original PEGASUS model was trained using a codebase based on TensorFlow. The experiments in this paper were run using a new codebase written with JAX (Bradbury et al., 2018) and Flax (Heek et al., 2020). PEGASUS-X$_{Base}$ and PEGASUS-X were trained by converting the weights from the TensorFlow checkpoint to a Flax checkpoint format, and then continuing with long input training.

| Cross-Attention | Model | arXiv | | GovReport | |
|---|---|---|---|---|---|
| | | R1 / R2 / RL | RG | R1 / R2 / RL | RG |
| Pretrained | Full | 47.7 / 20.4 / 28.6 | 30.3 | **61.3 / 29.4 / 30.8** | **38.1** |
| | Cross[0,2,4,6,8,10] | **48.1 / 20.4 / 28.6** | **30.4** | 61.0 / 29.0 / 30.7 | 37.9 |
| | Cross[0,6] | 47.1 / 19.8 / 28.1 | 29.7 | 60.4 / 28.1 / 29.7 | 36.9 |
| Converted | Cross[0,2,4,6,8,10] | 46.4 / 19.7 / 28.1 | 29.5 | 60.2 / 28.8 / 30.3 | 37.4 |
| | Cross[0,6] | 46.2 / 19.7 / 28.1 | 29.5 | 60.2 / 28.1 / 29.8 | 36.9 |

Table 13: Comparison of models pretrained with cross-attention for a subset of layers, and adapting a pretrained model by dropping cross-attention layers only during fine-tuning

| Model | CNN/DailyMail | | XSum | |
|---|---|---|---|---|
| | R1 / R2 / RLs | RG | R1 / R2 / RLs | RG |
| PEGASUS$_{Base}$ | 41.8 / 18.8 / 38.9 | 31.3 | 39.8 / 16.6 / 31.7 | 27.6 |
| PEGASUS$_{Base+}$ | 42.5 / 20.1 / 39.6 | 32.4 | 43.8 / 21.2 / 36.0 | 32.2 |
| PEGASUS-X$_{Base}$ | 42.5 / 20.1 / 39.6 | 32.4 | 42.9 / 20.1 / 35.0 | 31.2 |
| PEGASUS$_{Large}$ | **44.2 / 21.5 / 41.1** | **33.9** | **47.2 / 24.6 / 39.2** | **35.7** |
| PEGASUS-X | 43.4 / 21.2 / 40.6 | 33.5 | 45.8 / 22.8 / 37.6 | 34.0 |

Table 14: Comparison on short summarization tasks (Test sets)

| Dataset | Batch Size | Learning Rate | Num Steps | Max Input Tokens | Max Output Tokens | Beam Size | Beam Alpha |
|---|---|---|---|---|---|---|---|
| | | | PEGASUS-X$_{Base}$ | | | | |
| XSum | 64 | 8e-4 | 97.5K | 1024 | 128 | 4 | 0.8 |
| CNN/DailyMail | 64 | 8e-4 | 410K | 1024 | 128 | 4 | 0.8 |
| arXiv | 64 | 8e-4 | 92.5K | 16384 | 256 | 1 | 1 |
| Big Patent | 64 | 8e-4 | 272.5K | 16384 | 256 | 1 | 1 |
| PubMed | 64 | 8e-4 | 85K | 8096 | 256 | 1 | 1 |
| GovReport | 64 | 8e-4 | 40K | 12288 | 1024 | 2 | 1 |
| SummScreen | 64 | 8e-4 | 90K | 16384 | 256 | 1 | 1 |
| QMSum | 64 | 8e-4 | 7.5K | 16384 | 256 | 1 | 1 |
| | | | PEGASUS-X | | | | |
| XSum | 64 | 8e-4 | 5k | 1024 | 128 | 4 | 0.8 |
| CNN/DailyMail | 64 | 8e-4 | 7.5k | 1024 | 128 | 4 | 0.8 |
| arXiv | 64 | 8e-4 | 85k | 16384 | 256 | 1 | 1 |
| Big Patent | 64 | 8e-4 | 390k | 12192 | 256 | 1 | 1 |
| PubMed | 64 | 8e-4 | 47.5k | 12192 | 256 | 1 | 1 |
| GovReport | 64 | 8e-4 | 75K | 12288 | 1024 | 1 | 1 |
| SummScreen | 64 | 8e-4 | 40K | 12192 | 256 | 1 | 1 |
| QMSum | 64 | 8e-4 | 35K | 12192 | 256 | 1 | 1 |

Table 15: Hyperparameters for fine-tuning models