# OpenReview forum: "Investigating Efficiently Extending Transformers for Long Input Summarization"
_EMNLP/2023/Conference — EMNLP 2023 Main_

### Official Review · Reviewer_5wpq · 2023-08-04

**Soundness:** 5

**Excitement:**

4: Strong: This paper deepens the understanding of some phenomenon or lowers the barriers to an existing research direction.

**Paper Topic And Main Contributions:**

This paper optimizes the block attention-based efficient long-context Transformer model by extensively studying the effects of various design choices, such as the block size, the number of global tokens, the arrangement of blocks, the positional encoding, and the pre-training scheme. Part of their experimental results show that, in order to achieve better performance on long document summarization, it is crucial to have larger block sizes, more global tokens, staggered blocks. Based on their findings, they adapt the pre-trained encoder-decoder model PEGASUS to a long-context model named PEGASUS-X, which performs better on long document summarization datasets including GovReport and PubMed than existing pre-trained models. To sum up, their contributions are:
- They carry out an extensive study of design choices in a block attention-based efficient Transformer.
- They introduce a new long-context pre-trained Transformer, PEGASUS-X.

**Questions For The Authors:**

- Why do you choose to only conduct the study on long document summarization? It seems to me that this study can be beneficial to more tasks, such as long-form QA.

**Reasons To Accept:**

- They extensively study the effect of important design choices in a block attention-based efficient Transformer on long document summarization. In my opinion, it is of huge significance to conduct such a solid ablation study. Though they don't have a novel model design and one can probably find many existing models similar to the model they experiment with, prior work that introduces new models rarely investigate the effect of their designs with apple-to-apple comparisons, thus failing to truly reveal the effectiveness of those model design. For example, the difference in the pre-training data is a common confounding factor. This work is doing a pretty good job revealing the effects of those design choices.
- They also adapt the PEGASUS model to a long-context version, PEGASUS-X. It has the best performance on PubMed and GovReport.

**Reasons To Reject:**

- The domains of data used in their study are limited. It would be great to cover more difficult tasks such as meeting summarization and narrative summarization. Though SummScreen and QMSum are tested in the final experiment comparing PEGASUS-X with existing pre-trained models, having one of those datasets in the ablation study would make it more extensive.

**Reproducibility:**

5: Could easily reproduce the results.

**Reviewer Confidence:**

5: Positive that my evaluation is correct. I read the paper very carefully and I am very familiar with related work.

---

> ### Author Rebuttal · Authors · 2023-08-27
>
> Thank you for the kind feedback.
>
> We chose to study long-context models in the setting of summarization tasks as it is a well-studied task with established datasets used in other academic works. Focusing on a single category of tasks with well-established (if imperfect) metrics allowed us to experiment more extensively with architectural and training configurations, without needing to juggle too many different evaluation settings or choices (e.g. evaluation of generated responses for QA).
>
> We agree that summarization is limited in scope compared to the broader range of long-context tasks, and that even among summarization tasks there are many sub-genres and domains of data. This work can certainly be extended to cover broader domains of data and tasks, where different model configurations may be advantaged or disadvantaged in different settings. We see this work as an attempt to carefully and meticulously study the model and training configurations within a fixed and well-established if slightly narrow scope.

---

### Official Review · Reviewer_8ipF · 2023-08-05

**Soundness:** 4

**Excitement:**

3: Ambivalent: It has merits (e.g., it reports state-of-the-art results, the idea is nice), but there are key weaknesses (e.g., it describes incremental work), and it can significantly benefit from another round of revision. However, I won't object to accepting it if my co-reviewers champion it.

**Paper Topic And Main Contributions:**

The paper studies long input summarization tasks and introduces an extension model (called PEGASUS-X) to the pretrained PEGASUS model. PEGASUS-X has a staggered and block-local Transformer architecture with global encoder tokens and it provides a better trade-off between task performance and efficiency. PEGASUS-X supports input tokens with 16K and shows strong task performance compared to baseline models.


The paper has several contributions: (1) a systematic study of the model architectures for long input summarization, including efficient encoders, local and global block Transformers, and their combinations (2) a simple and effective architecture extension based on the findings from the study, strong evaluation results for both long and short summarization tasks.


**Reasons To Accept:**

The paper presents a strong long input Transformer model for summarization tasks. The architecture design choices are solid and well-studied.

**Reasons To Reject:**

The empirical effectiveness seems to be insignificant to previous models like BART-LS which has even fewer parameters. For example, PEGASUS-X in Table 6 and 7 both show marginal (<1%) improvements over BART-LS. It is also unclear whether PEGASUS-X is faster than BART-LS or not.

**Reproducibility:**

4: Could mostly reproduce the results, but there may be some variation because of sample variance or minor variations in their interpretation of the protocol or method.

**Reviewer Confidence:**

3: Pretty sure, but there's a chance I missed something. Although I have a good feel for this area in general, I did not carefully check the paper's details, e.g., the math, experimental design, or novelty.

---

> ### Author Rebuttal · Authors · 2023-08-27
>
> Thank you for the kind feedback.
>
> Regarding BART-LS: we mentioned in the related work that BART-LS is work concurrent to this manuscript. More broadly, BART-LS investigates a similar set of methods and settle on a similarly configuration, using a staggered block-local attention and extended pretraining for the best performing version. BART-LS is in between PEGASUS-X-base and PEGASUS-X-large in the number of parameters, and is also trained on a different set of data. Hence, while the numerical results are not directly comparable (in terms of extrapolating conclusions on architecture), we find that the good performance of BART-LS is consistent with the overall findings of our work.

---

### Official Review · Reviewer_fXYW · 2023-08-19

**Typos Grammar Style And Presentation Improvements:** Nothing comes to my notice.
**Soundness:** 4

**Excitement:**

4: Strong: This paper deepens the understanding of some phenomenon or lowers the barriers to an existing research direction.

**Missing References:**

Nothing comes to my notice.

**Paper Topic And Main Contributions:**

This paper addresses the problem of effectively and efficiently handling long inputs in summarization tasks. It investigates various improvements to Transformer models to tackle the challenges posed by long-document summarization. The paper proposes PEGASUS-X, an extended version of PEGASUS, with a modified architecture and additional long-sequence pretraining. The goal is to achieve state-of-the-art performance on long-input summarization tasks while being computationally efficient.

**Questions For The Authors:**

A. What is the impact of different choices of global token embeddings on the performance of PEGASUS-X for long-document summarization tasks?
B. How does the performance of PEGASUS-X compare to other models for long-document summarization in terms of computational resources required for training and inference?

**Reasons To Accept:**

- The paper conducts extensive ablation experiments and empirical investigations to explore various improvements to Transformer models for long-document summarization tasks.
-  PEGASUS-X achieves state-of-the-art performance on long-input summarization tasks in two datasets: GovReport and PubMed.
- The paper addresses the computational efficiency of long-document summarization models. It proposes a block-wise local Transformer architecture with staggered blocks and global tokens, striking a balance between performance and memory efficiency.

**Reasons To Reject:**

Nothing comes to my notice.

**Reproducibility:**

3: Could reproduce the results with some difficulty. The settings of parameters are underspecified or subjectively determined; the training/evaluation data are not widely available.

**Reviewer Confidence:**

3: Pretty sure, but there's a chance I missed something. Although I have a good feel for this area in general, I did not carefully check the paper's details, e.g., the math, experimental design, or novelty.

---

> ### Author Rebuttal · Authors · 2023-08-27
>
> Thank you for the kind feedback.
>
> A) We show in Section 3.3 / Table 3 that having more global tokens generally leads to better performance, albeit at a memory and computational cost.
> B) At inference time, the computational cost should be similar to other models with Global+Local layers (e.g. Longformer), with the exact performance depending on the optimizations used (e.g. Longformer uses custom CUDA kernels). For training, the primary cost comes from more long-context pre-training, but we feel our investigation (Section 3.6 / Table 4) provides a good answer to the trade-off in the short vs. long table.

---

### Meta-Review · Area_Chair_nJ3d · 2023-09-18

**Recommendation:** 5

**Metareview:**

This paper addresses the challenge of effectively handling long inputs in summarization tasks by proposing PEGASUS-X, an extended version of PEGASUS, with a modified architecture and additional long-sequence pretraining. PEGASUS-X aims to perform state-of-the-art long-input summarization tasks while maintaining computational efficiency. It introduces a block-wise local Transformer architecture with staggered blocks and global tokens to balance performance and memory efficiency. PEGASUS-X demonstrates strong performance on datasets like GovReport and PubMed. However, the paper's limitation lies in its domain coverage, as it primarily focuses on specific datasets and broader tasks like meeting and narrative summarization tasks are not included in the study. Also,  the paper's empirical effectiveness compared to BART-LS shows marginal improvements.

---

### Decision · Program_Chairs · 2023-10-07

**Decision:**

Accept-Main

**Comment:**

This paper addresses the challenge of effectively handling long inputs in summarization tasks by proposing PEGASUS-X, an extended version of PEGASUS, with a modified architecture and additional long-sequence pretraining. PEGASUS-X aims to perform state-of-the-art long-input summarization tasks while maintaining computational efficiency. It introduces a block-wise local Transformer architecture with staggered blocks and global tokens to balance performance and memory efficiency. PEGASUS-X demonstrates strong performance on datasets like GovReport and PubMed. However, the paper's limitation lies in its domain coverage, as it primarily focuses on specific datasets and broader tasks like meeting and narrative summarization tasks are not included in the study. Also,  the paper's empirical effectiveness compared to BART-LS shows marginal improvements.